



# Application of an O-ring pinch device as a constant pressure inlet (CPI) for airborne sampling

Sergej Molleker[1], Frank Helleis[1], Thomas Klimach[1], Oliver Appel[1,2], Hans-Christian Clemen[1], Antonis Dragoneas[1,2], Christian Gurk[1], Andreas Hünig[1,2], Franziska Köllner[1], Florian Rubach[1,3], Christiane Schulz[1,3], Johannes Schneider[1], and Stephan Borrmann[1,2]

[1]Max Planck Institute for Chemistry, Mainz, Germany
[2]Institute for Atmospheric Physics, Johannes Gutenberg University, Mainz, Germany
[3]Leibniz Institute for Tropospheric Research, Leipzig, Germany

**Correspondence:** Sergej Molleker (s.molleker@mpic.de), Frank Helleis (frank.helleis@mpic.de)

**Abstract.** We present a novel and compact design of a constant pressure inlet (CPI) developed for use in airborne aerosol mass spectrometry. In particular the inlet system is optimized for aerodynamic lenses commonly used in aerosol mass spectrometers, where efficient focusing of aerosol particles into a vacuum chamber requires a precisely controlled lens pressure, typically of a few hPa. The CPI device can also be used in gas phase sampling instruments in a large range of altitude and inlet pressure. The constant pressure is achieved by changing the inner diameter of a properly scaled O-ring that acts as a critical orifice. The CPI control keeps air pressure and thereby mass flow rate ($\approx 0.1$ l/min) upstream of an aerodynamic lens constant, deviating at most by only $\pm 2\%$ from a pre-set value. In our setup a pressure sensor downstream of the O-ring controls the pinch mechanism via a feedback loop and setpoint conditions are reached within seconds. The device was implemented in a few instruments, which were successfully operated on different research aircraft covering a wide range of ambient pressures, from sea level up to about 55 hPa. Details of operation and quality of aerosol particle transmission are evaluated by laboratory experiments and in-flight data with a single particle mass spectrometer.

## 1 Introduction

There is a growing field of airborne atmospheric measurements of aerosol particles performed in earth science applications (Fuzzi et al., 2015). It is of major interest to study the chemical composition of aerosol particles which, among other methods, can be investigated by mass spectrometry. Improvements regarding electronics, size and weight (still in the range of 75–250 kg) enabled online in-situ mass spectrometry on many airborne platforms covering a wide altitude range. Those instruments, either of single particle laser ablation type (Murphy and Thomson, 1995; Zelenyuk et al., 2015) or flash vaporization (AMS-Instrument, Drewnick et al., 2005; Canagaratna et al., 2007) commonly use an aerodynamic lens (Liu et al., 1995; Kamphus et al., 2008) to permit and focus aerosol particles into the vacuum chamber. Aerodynamic lenses are designed for a narrow range of lens pressure ($P_{lens}$), which depends on the desired particles size range. Before constant pressure devices emerged, fixed critical orifices were used, appropriately sized in diameter either for sea level pressure or a pressure at a certain flight altitude of interest. For deviating inlet pressures this requires correction of the flow rate and size-dependent particle transmission. Such post-flight data evaluation is based on laboratory calibrations (Bahreini et al., 2003). This approach introduces uncertainties and is not optimal for quantitatively measuring instruments, for instance of AMS-type, which is already relying on many other calibrations, such as for collection efficiency, ionization efficiency, lens transmission, focusing and alignment. Also, the time-of-flight method for obtaining individual vacuum aerodynamic sizes of aerosol particles, as integrated on some instruments, relies on a known and preferably constant lens pressure. Hence, it is desirable to actively maintain the lens pressure at a





constant value independent of the ambient/inlet pressure and thus on flight altitude. An aircraft configuration of a constant pressure inlet is indicated in Fig. 1. One alternative solution is to introduce an intermediate pressure volume downstream of a critical orifice. This volume is differentially pumped to keep it and the lens at constant pressure. In the initial demonstration of such a design (Bahreini et al., 2008) a minimum pressure altitude of 600 hPa was considered. In following airborne applications,

successful measurements at altitudes up to 12 km (Schmale et al., 2010) and up to 7.7 km (Pratt et al., 2010) were reported. The advantages and limitations of such differentially pumped volume systems are discussed in Section 6. In our different approach, a very compact constant pressure inlet (CPI) setup, without additional pumping and a bypass flow was developed and flight proven for single particle and flash-vaporization type instruments, at altitudes from ground level up to 20 km altitude (e.g. Köllner et al., 2017; Schulz et al., 2018; Höpfner et al., 2019; Dragoneas et al., 2020; Schneider et al., 2020).

## 2 Principle of operation and technical description

The idea behind our CPI design is a realization of a critical orifice with a variable diameter of the opening's cross section. The constant pressure inside the aerodynamic lens implies a constant mass flow rate of air (assuming constant air temperature). Considering that the mass flow rate through a critical orifice is proportional to its cross-section area and upstream pressure, it follows: to maintain a constant mass flow over a wide range of ambient pressure, for instance between 1000 hPa and 60 hPa,

the circular cross-sectional area of the orifice has to change by a factor of approximately 16.7 (1000/60). This corresponds to a factor of about 4.1 of change in diameter. Aerosol mass spectrometers typically require/use a fixed orifice of 0.1 mm diameter at sea level pressure (e.g. Jayne et al., 2000; Drewnick et al., 2005; Zelenyuk et al., 2009). Hence, the inner diameter of the relaxed O-Ring has to be 0.4 mm to 0.5 mm, shrinkable by pinching down to 0.1 mm. The initial design started with commercial O-rings of $0.8 \times 2$ mm (Inner Diameter $\times$ Cross Section) in dimension, acquired from "Dichtelemente arcus GmbH". Models like

FKM/FPM75 or MVQ70 (abbreviation indicates elastomer type and "'shore hardness'") were tested for particles measurements (black and red variants in Fig. 2). In a next step, for improving the particle transmission, custom made O-rings with dimensions of $0.5 \times 2.1$ mm and $0.4 \times 2.15$ mm were produced (for the same mechanical setup that fixes and holds the O-ring). Two-component polymerizing silicon rubber of the brand "Zhermack" with shore-hardness of 50 (blue O-rings in Fig. 2) emerged to be better suitable material regarding good orifice shape during compression/pinching as compared to even softer elastomers.

The shape of the pinched orifice is substantial to avoid significant particle losses, as discussed in the following section. The outer diameter of the pinch O-ring i.e. tube's cross section was chosen to be relatively large as the larger bulk of the O-ring rim allows for more compression and shrinking of its inner opening.

The pinch O-ring is mounted inside a cylindrical housing on a moving/tilting lever, which pushes the O-ring against the surface of a counterpart. In our case, this counterpart is a plate welded on top of a 1/4" plug valve block (see principal

operation drawing in Fig. 3). To minimize the probability of turbulence the variable O-ring's orifice diameter, the openings of the housing, both upstream and downstream of the O-ring ($D_{bore}$), are designed as large as possible. Yet the openings must be small enough to ensure air-tight connection and sufficient mechanical support surface for the O-ring. In the presented version, both diameters ($D_{bore}$) of 1 mm are equal. Initially, the length of both bore holes in the described setup was 1.5 mm, but was further reduced to 0.5 or 1 mm in newer designs. Downstream of the O-ring the tube expands conically in 45° to 4.2 mm (i.e.

the clear diameter of the 1/4" plug valve in Fig. 3). In the most recent mechanical design (not use for characterization here), the downstream tube expands directly from the O-ring from 2 mm to 39 mm at an angle of 5.6° conically. This is to further reduce radial flow velocities due to jet expansion at the exit of the critical orifice (Hwang et al., 2015).

The sample tube upstream of $D_{bore}$-tube has an inner diameter (DST) of 2.05 mm (1/8" OD tube). The tilting part of the O-ring's housing (lever) is pivoted and its axis is designed 10 mm away to ensure small deviation from parallel configuration of the

40 mechanism compressing the O-ring. Tilting along the direction of the curved arrow squeezes the O-ring and its inner opening. The pinching movement is driven by a motor, which is pulling the lever via a spindle. The opposite, relaxation movement is supported by the spring force of the O-ring and an additional spring.

A custom-made electronic control unit drives the motor until a pre-set lens pressure is reached. The control loop uses the output of a lens pressure sensor as feedback. In the presented example (ERICA instrument, Hünig et al., 2020), good results

were obtained with an absolute pressure sensor (Pfeiffer CMR373-Model), with a range of 0–10 hPa and 0.15 % accuracy. The feedback of the control unit was configured only with the integrative component (PID terminology, Proportional-Integrative-Differential): the speed of the servo motor is proportional to the deviation from the set lens pressure. Good control stability of the lens pressure was demonstrated during the duration of longer flights on several aircraft (G550 HALO, Polar 6, M-55 Geophysica, DC-8, DLR Falcon). Pressure deviations were within $\pm 2$ %, including ascent or descent rates of up to 30 m/s

(StratoClim test campaign with M-55). Figure 4 exemplifies the CPI-controlled lens pressure during one of the flights where altitudes of up to 20 km were reached. As shown in the graph, the inlet tube pressure decreased down to 64 hPa, a value close to the estimated maximum possible aperture for the 0.5 mm inner diameter O-ring. At this point it can be added that at minimum



inlet pressure some of the pre-pinching is reserved to ensure an air tight connection of the O-ring's outer surfaces with respect to the inlet air.

The plug valve, as indicated in the Fig. 3, is built as part of the CPI system, but is independent from the feedback control. The position of the lens pressure sensor between the O-ring and the valve adds safety for the vacuum system during opening (and closing) of the inlet line in the following way: after closing or at closed position of the valve a rise in the lens pressure forces the feedback control to shrink the O-ring to its (set) minimum. Therefore, at the moment of opening of the valve, only a low and safe flow rate can enter the vacuum chamber before the control starts to increase the O-rings aperture to reach a set lens pressure within about 15–20 seconds. The pinching movement is limited on both sides by a signal of an optical sensor, which is sensing the distance to the moving part. This provision prevents overloading of the motor at maximum pinching and as a backup prevents the mechanism from opening too far, which might result in leakage of ambient (laboratory/cabin) air into the sample line.

## 3 Visual inspection of the pinched O-ring

With the aerosol mass spectrometry, as the main application in mind, the CPI performance has to be assessed by the quality of particle transmission in combination with an aerodynamic lens. As shown in Fig. 4, the performance regarding lens pressure stability meet the requirements. Initial particle transmission measurements with the commercial 0.8 mm-inner-diameter O-ring revealed high particles losses. The highest losses of up to about 80 % (for 350 nm sized ammonium nitrate particles) were observed for a high degree of pinching at sea level pressure, while for lower inlet pressure (i.e. less pinching), the particle transmission improved significantly and yielded results comparable with those of fixed orifice configuration. The transmission losses were preliminary evaluated by comparing the overall/bulk transmission and ionization efficiency of flash-vaporization MS (AMS-instrument) versus identical AMS configurations equipped with fixed orifice inlets. Additionally, comparisons were performed by observing optically detected counts of a single particle laser ablation instrument versus external particle counting instruments. This high particle transmission loss behaviour could be explained by further investigation of the shape of the O-ring's aperture during pinching.

For this purpose, photographs such as those illustrated in Fig. 5, were taken with a long working distance (70 mm) microscope looking through the sampling tube and the O-ring pinched inside the CPI device. For sea level pressure, whereby the state of pinching was known by the previously recorded signal of the distance sensor, the 0.8 mm O-ring underwent a deformation into a narrow-slit shape (upper left panel in Fig. 5a). At lower inlet pressure and consequently less pinching, the same O-ring regains its round shape. Here, the visible illuminated aperture area roughly corresponds to the fixed orifice area needed for the given mass flow. The tested inlet pressure limits are given in the legend for some of the photographs. Obviously, the 0.8 mm diameter is unnecessarily large and some of the compression can be avoided by starting with a smaller inner diameter. Additionally, a softer elastomer material was expected to be better for keeping the O-ring's aperture round. Therefore, 0.5 mm and 0.4 mm inner diameter O-rings were produced in machined molds using off-the-shelf two-component silicon rubber. Two different materials with shore hardness of 50 and 22 were tested, with the latter showing less suitable properties. Figure 5b illustrates the behaviour of the "0.5 mm O-ring" (shore 50). When pinching corresponding to the sea level pressure, the aperture resembles a triangle. Still, successive transmission efficiency measurements gave values similar to those measured by a fixed orifice setup. The same good behaviour was exhibited by the 0.4 mm inner diameter version, however, in order to keep the following calibrations more general, a 0.5 mm-O-ring was chosen for use in all of the instruments of the research group. Furthermore, the larger diameter is suitable for the maximum expected flight altitude.

Laboratory tests with that configuration preceded the application of the system in stratospheric flights with inlet pressure down to 55–60 hPa (lower right picture in Fig. 5b). Notably, the lit area or the aperture of the 1000 hPa-photograph (Fig. 5b, top left) has about the area equivalent to a 0.1 mm- diameter orifice required for the same lens pressure. A comparison to the 0.8 mm O-ring at sea level pressure (Fig. 5a, top left) shows that, while the 0.8 mm O-ring controls the same lens pressure and hence the same airflow, the aperture cross section must be similar, but is hidden from the front view. Supposedly, the "slit aperture" has a warped shape where most of the high particle loss occurred.

## 4 Methods: Laboratory characterization of the particle transmission

After optimizing the O-ring's pinching properties, particle size dependent transmission measurements were performed with a laboratory setup. First experiments focused on conditions at sea level pressure, where according to first experience, high particle losses can occur due to deviations from circular shape of the aperture and its minimum size. The work presented below utilizes an optical detection unit of the aerosol mass spectrometer ERICA (Hünig et al., 2020), which is a combination of a single particle laser ablation ToF-MS and a flash-vaporization ToF-MS. The aerodynamic lens used here was the intermediate





pressure lens (IPL, Peck et al., 2016) optimized for PM2.5, a model from Aerodyne Research, Inc. In practice, the CPI's particle transmission can be tested only in combination with an aerodynamic lens, meaning that separate contributions of the individual components to particle losses are not directly accessible. The approach presented here is an absolute particle transmission measurement conducted in the same configuration for the CPI as for the fixed orifice setup. The lens pressure for CPI operation was set to 4.53 hPa and a slightly different lens pressure of 4.41 hPa was measured behind a 100 µm fixed orifice. In both cases, sample flow rates were measured and taken into account (CPI: 1.48 cm$^3$ s$^{-1}$ and fixed: 1.31 cm$^3$ s$^{-1}$). Particle size dependent transmission experiments were performed using polystyrene latex (PSL) beads, generated from a water solution (atomizer), with diameters between 200 nm and 5 µm. The number concentrations were referenced by a CPC (Grimm CPC Model 5.403) for sizes up to 800 nm or by an OPC (Grimm SkyOPC, Series 1.129, Bundke et al., 2015) for PSL sizes above or equal to 1000 nm.

The applied optical detection unit and the measurement technique are described in more detail in Hünig et al. (2020), but are briefly introduced in the following paragraph. In the ERICA instrument, the particle beam directed by the aerodynamic lens intercepts a perpendicular cw-laser beam (UV light at 405 nm) at a distance of 59 mm. Here, the laser focus is adjusted to coincide with the focal point of the light collecting elliptical mirror. The light scattered by individual aerosol particles is detected by a photomultiplier tube situated at the other focal point of the elliptical mirror. To obtain the total particle transmission of the aerodynamic lens, i.e. not to account for parts of the particle beam lost outside the laser/mirror foci below detection threshold, a method described in the following is applied. The aerodynamic lens is mechanically stepped to move the narrow particle beam like a scan across the detection laser beam. Due to a relatively moderate focusing of the detection laser ($\approx$7 mm Rayleigh range), the detection region along the laser beam axis is assumed to extend an order of magnitude more than across the beam (beam waist $\approx$60 µm). Therefore, scanning in one dimension is considered sufficient for PSL diameters larger than 200 nm. For particle sizes below 150–200 nm the optical detection region decreases steeply yielding only 1–2 % detection efficiency at 100 nm. This prevents optical detection for particle transmission studies. From the measured detection efficiency profile (DE) as a function of lens position (x coordinate), the total transmission efficiency (TE) can be derived. For doing so, a simple but well performing model was assumed by approximating the particle beam with a two-dimensional-Gaussian profile standard deviation $\sigma$ (Klimach, 2012). Since the scattered light pulses are recorded only above a certain intensity threshold, it is approximated by a rectangular top-hat function of the width $2r_L$. The measured detection efficiency corresponds to the convolution integral of the both functions and the result of the integral is given in Eq. (1):

$$DE(x) = 0.5\, TE \left( erf\left(\frac{x + r_L - x_0}{\sqrt{2}\sigma}\right) - erf\left(\frac{x - r_L - x_0}{\sqrt{2}\sigma}\right) \right) \tag{1}$$

Two examples of detection efficiency scans are shown in Fig. 6 together with the fitted functions (red) as described by Eq. (1). The error bars are based on the Poisson counting uncertainty of the detection unit and the external CPC or OPC instruments. Fit parameters are $TE$, $r_L$, $x_0$ and $\sigma$, with $TE$ being the main result for lens transmission. If the particle beam is narrower than the optical detection region, the measured profile shows a pronounced plateau at a magnitude equal to the transmission efficiency (Fig. 6, CPI example). For scans with the particle beam width being comparable or larger than the detection width, the obtained $TE$ differs from maximum detection efficiency. In most detection-scans the maximum detection efficiency and the fitted $TE$ differed by less than 5 %.

Intuitively, the CPI performance is expected to improve with decreasing inlet pressure due to relaxation of the O-ring, which would regain a larger and smoother round shape. On the other hand, a decrease of inlet pressure enhances the effects of other factors like the increasing flow speed and growing Cunningham slip correction factor $C$ (due to decrease in air density the flow speeds up for keeping the mass flow constant). Both parameters lead to larger Stokes number $St$ as a measure of particle impaction losses. The effects on the Stokes number can be seen in Eq. (2), which is valid for the front side of the flat round orifice (Lee et al., 1993; Bahreini et al., 2008),

$$St = \frac{\rho_p D_p C U_0}{18\,\mu\,D_0} \tag{2}$$

with particle density $\rho_p$, particle diameter $D_p$, Cunningham slip correction factor $C$, sample air velocity upstream of the orifice $U_0$, dynamic air viscosity $\mu$ and orifice diameter $D_0$. The O-ring geometry represents a smoother obstacle for the air stream, such that Eq. (2) only provides an (upper) approximation for the Stokes number, which still is useful for dimensional analyses. By replacing all variables dependent on inlet pressure, such as orifice diameter, air velocity and slip correction, Eq. (2) is evaluated in the pressure range from 30 to 1020 hPa (Fig. 7). A few examples for different particles sizes, nominally of density equal to 1 g cm$^{-3}$, calculated for a 1/8" (OD) upstream sample tube are shown. Since the Stokes number is proportional to particle density results for higher density can be easily estimated. The plot shows that for inlet pressures from sea level





down to about 300 hPa particles losses for submicron particles should be low ($St < 0.1$, including a margin for higher particle density) and hardly distinguishable from the transmission of the aerodynamic lens. For sea level pressure this is confirmed by the laboratory measurement (Fig. 8). Towards stratospheric inlet pressures (150–60 hPa) the Stokes number increases more rapidly and significant particle loss at diameters of about 1 µm and above can be expected.

To obtain particle transmission at lower inlet pressure, i.e. to simulate flight conditions, a differentially pumped volume downstream of a 250 µm diameter orifice was set up in the laboratory. Downstream of the low pressure volume, sampling lines to the CPI/mass spectrometer and a SkyOPC were connected via a Y-manifold. Measurements with the CPI system were performed at a few different values of inlet pressure, reducing pressure in steps by about factor two such that pressures of 380, 250, 125 and 65 hPa were tested. The highest pressure in this setup/procedure was chosen below 500 hPa, e.g. 380 hPa, with the aim to be also included in critical flow condition ($P_{downstream} < 0.5\,P_{upstream}$) at the pressure reducing orifice. This way the sampled flow and hence particle losses from the particle generation to the pumped volume can be assumed as being nearly constant (for all tested values of inlet pressure). Additionally, care was taken to run and observe the particle generation at constant concentration (well within 2 %) during an experiment. The latter effort was done due to a lack of a reliable particle concentration reference within the low-pressure setup: although the SkyOPC is designed for measurements at low inlet pressure (specified down to 125 hPa), the changes in measured particle concentration appeared to be higher (and unknown) than would be acceptable for use as a reference for CPI characterization. Only the (absolute) transmission value at 380 hPa was obtained by referencing photomultiplier counts to this external SkyOPC. From there on for every particle size, inlet pressure was varied in steps down to 65 hPa and raised back again to 380 hPa. At every pressure step, time was given for the CPI pinching mechanism to adjust, and then particle counts with statistically sufficient time were recorded by the mass spectrometer's optical detection unit (at the maximum detection lens position). Transmission values were obtained as relative changes to the starting point.

## 5  Particles transmission results

Detection scans were performed for various PSL particles sizes at ground pressure (laboratory at 130 m above sea level). The resulting transmission efficiencies for both setups, the CPI O-ring and a fixed 100 µm orifice setup, are summarized for comparison in Fig. 8. Both lines (blue and black) are shown with filled data markers. The following contributions to the error bars are included: uncertainty of the fitted TE-parameter (1-5 %), particle counting uncertainties (<2 %), sample flow uncertainty (3 %), and for sizes above 1 µm the uncertainty of the tube loss correction in the tubing to the OPC (e.g. 44 % for 5 µm particle diameter). The transmission efficiency is mainly defined by the performance of the lens and compares well to results of the same lens design published in Xu et al. (2017). The transmission values of the CPI are within uncertainties very similar to those of a fixed orifice. As expected from a simple model for Stokes number dependence (Eq. 2, Fig. 7) the decreasing inlet pressure leads to a reduction of particle transmission with increasing particles size. The same transmission data is plotted (for tested particles sizes) as a function of pressure in Fig. 9. The results in Fig. 9 also reflect the measurement procedure, for which the inlet pressure was varied in steps for the same particles size. The general behavior confirms the theoretic expectation of a decreasing particle transmission with decreasing inlet pressure. The size of 200 nm seems to be not affected yet by the increasing Stokes number. Some steps down in pressure show an opposite effect, i.e. a slight increase in particle transmission. The latter can most likely be attributed to the effect of an improving O-ring opening geometry, transforming from a shape resembling a triangle to a circular aperture.

## 6  Summary and discussion

A successful operation of the presented pinch O-ring "constant pressure inlet" device has been demonstrated by means of suitable laboratory characterization experiments. An inlet pressure range between sea level and down to 65 hPa was covered with a single properly dimensioned O-ring. The CPI setup is compact: it adds only about 6 cm to the length on top and along of the aerodynamic lens. Also, good stability with a deviation less than 1.2 % from the pre-set lens pressure (2.5 hPa to 5 hPa, depending on the instrument and aerodynamic lens type) was demonstrated during research flights (G550 HALO, M-55 Geophysica, DC-8). Furthermore, the CPI device was also applied in a gas phase sampling instrument, a proton transfer reaction mass spectrometer (PTR-MS Jordan et al., 2009), where a "drift tube" in the inlet path requires a constant pressure of 2.2 hPa.

Results of particle transmission characterized in laboratory show that for inlet pressure down to 250 hPa, which is within the maximum altitude of most passenger type research aircraft (e.g. ≈195 hPa ambient pressure at 12 km altitude and ≈60 hPa of dynamic pressure at 210 m/s flight speed), transmission of submicron aerosol particles remains larger than 80 % and shows small differences to transmission at ground level. For a higher flight altitude and aerodynamic particle diameters larger than 0.6–0.8 µm, corrections should be applied. Nevertheless, even at the lowest designed and experienced inlet pressure of 65 hPa





the CPI's particle transmission performance can be well justified for stratospheric aerosol particle sizes below $0.6\,\mu m$. In cases of higher optically sized (even mean volume) diameters, e.g. of fresh volcanic origin at the stratospheric altitudes (Wilson et al., 1993), the evaporation of the water content of particle types such as sulphuric acid-water solutions in the inlet tubing, would result in smaller particles' aerodynamic diameters before arriving at the CPI. This would effectively improve the particle transmission through the CPI, if one were judging by the ambient optical sizes.

Nevertheless, particle losses considered in this publication also occur at alternative pressure reducing inlet devices, like the setup with the fixed diameter orifice introduced in Bahreini et al. (2008), where a differentially pumped volume at an intermediate constant pressure is utilized as "representative" pressure. The O-ring based setup avoids the additional critical orifice (prone to particle losses) needed to create such intermediate pressure step and eliminates the additional time spent by trace gases exposed to additional surfaces. Furthermore, in the case of a minimum inlet pressure of $65\,hPa$, the pressure at the intermediate volume has to be set at about $30\,hPa$, which implies even higher Stokes numbers and particle losses at the constant diameter orifice upstream of the lens. In future designs the geometry for the flow around the O-ring can be further optimized (e.g. small angle conical diffusors on both sides) to further improve the particle transmission of the CPI.

*Author contributions.* S.M. conducted laboratory characterization and field testing, designed and produced custom-made O-rings, made all figures, wrote the manuscript with all the authors contributions and comments, F.H. suggested and developed the CPI system, T.K. developed hardware, A.D. and O.A. developed data acquisition software and characterization methods, C.G. developed control electronics; O.A., H.-C.C., A.D., A.H., F.K., and C.S. were involved in laboratory and field testing and data analysis, F.R. contributed to hardware development and its testing, J.S. conducted first field testing and initiated laboratory characterization, S.B. was involved in the initial designs and the manuscript drafting, all co-authors commented on the manuscript.

*Competing interests.* The authors declare no competing interests.

*Acknowledgements.* We are grateful for the work done by the mechanical and electronic workshops of the Max Planck Institute for Chemistry and the mechanical workshop of the Institute for Atmospheric Physics of the University of Mainz. We also appreciate the work of many people involved in the organization and conductance of several aircraft field campaigns where the CPI device was operated and tested. The characterization work was accomplished on the ERICA mass spectrometer, financed by the European Research Council ERC Advanced Grant of S. Borrmann (EXCATRO project, Grant No. 321040). The work was supported by the Max Planck Society. We acknowledge the support of the StratoClim field campaign (with the maximum tested flight altitude), financed by the European Community's Seventh Framework Programme (FP7/2007 - 2013, Grant No. 603557). Further supporters are the German Federal Ministry of Education and Research (BMBF, Joint Project SPITFIRE (01LG1205A) and ROMIC-Initiative (ROle of the MIddle atmosphere in Climate)) and the DFG (grant SCHN 1138/2-2, BO 1829/9-1 and WI 1449/24-1).

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




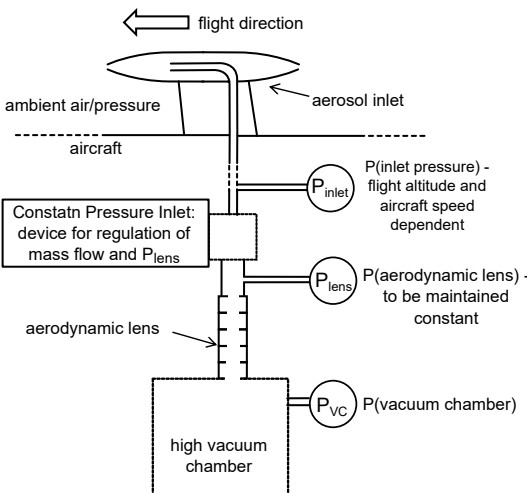

**Figure 1.** The position of a constant pressure inlet (CPI) device in the sampling path is indicated in the drawing. The CPI is located upstream of an aerodynamic lens. Bypass flow or manifolds to other instruments are omitted. The task of the CPI device is to keep the pressure at the aerodynamic lens ($P_{lens}$) constant, regardless of changes in the inlet pressure ($P_{inlet}$).

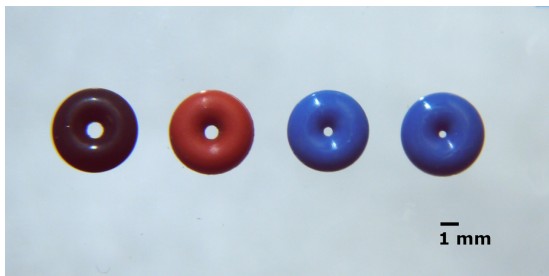

**Figure 2.** Photographs of pinch O-rings used during development of CPI. The black and red O-rings (left) are the initially adopted commercially available O-rings of $0.8 \times 2.0$ mm dimensions, with shore hardness of 75 (FPM) and 70 (MVQ), respectively. The blue O-Rings (right) with 0.5 mm and 0.4 mm inner diameters were produced in the laboratory out of two-component silicone rubber with shore hardness of 50. The 0.5 mm inner diameter O-ring was chosen as the optimal solution for the flight altitude of up to 20 km (minimum inlet pressure) and lens pressures used in the instruments of the research group. On the right hand side a 1 mm scale is shown.





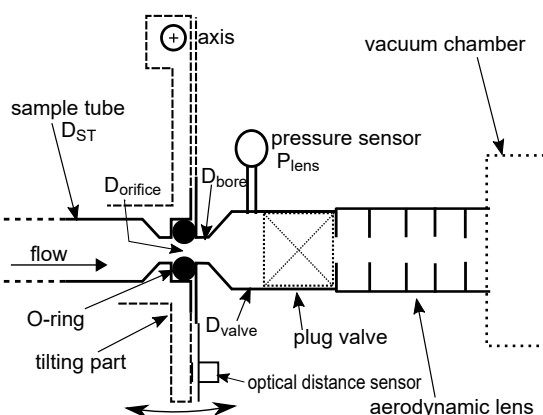

**Figure 3.** Principle drawing of the CPI design. The tilting part can be moved by a motor pulling it over a spindle mechanism. The controller uses the output of a pressure sensor downstream of the O-ring ($P_{lens}$), as indicated in the drawing. An additional optical distance sensor between the tilting part and the valve block is used to ensure safe stop limits for the movement. Relevant inner diameters for sample flow are denoted with $D_{ST}$ (2 mm), $D_{bore}$ (1 mm), $D_{orifice}$ (variable), $D_{valve}$ (4.2 mm).

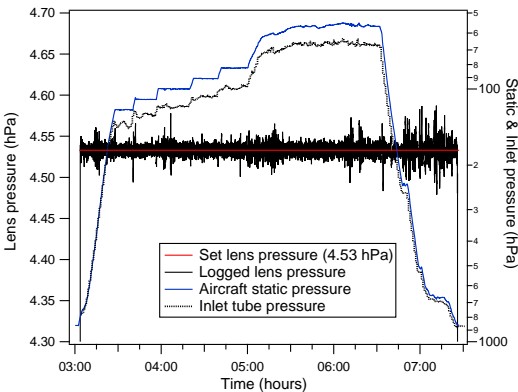

**Figure 4.** Stable CPI operation in the ERICA instrument is demonstrated by the recorded lens pressure aboard M-55 Geophysica aircraft (StratoClim Project, flight on 27. August 2017). The maximum deviation of the lens pressure from the set value is 1.2 %. The flight was chosen for its high maximum altitude (GPS) of 20.45 km with minimum inlet pressure of 64 hPa and ambient pressure of 55 hPa, which differ in dynamic pressure.





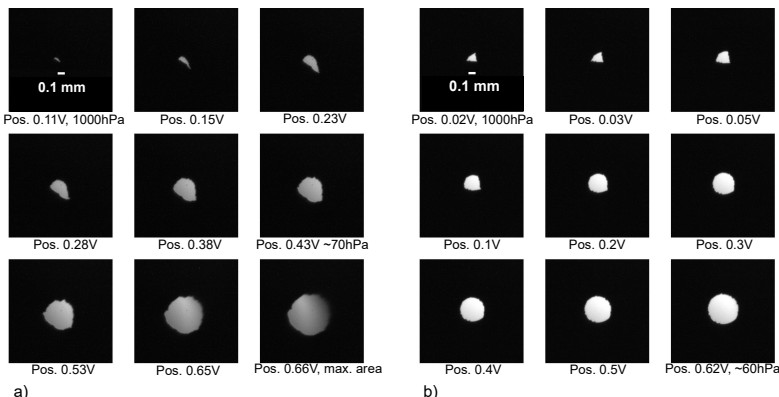

**Figure 5.** Microscope photographs of the O-ring aperture inside a backlit CPI device at different levels of pinching. The amount of pinching is recorded by a voltage output (photograph's captions) of an optical distance sensor installed on the CPI's mechanics (Fig. 3). From additionally logged inlet pressure during flight or laboratory experiments, the pinching position can be referred to a given inlet pressure. Exemplarily, pressure values of the desired limits (1000 hPa to 60 hPa or 70 hPa) are denoted at corresponding photographs. Panel a) shows the initial setup with a commercial O-ring having dimensions of $0.8 \times 2$ mm. As seen in the minuscule (backlit) aperture area for sea level pressure (1000 hPa), "warping" of the opening's shape and thus of the sample flow path can lead to high particles losses. Panel b) shows the custom-made produced optimized O-ring design with an inner diameter of 0.5 mm suitable for the lowest inlet pressure required.

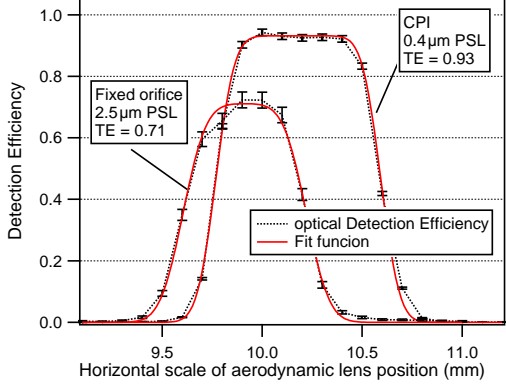

**Figure 6.** Two examples of detection efficiency measured by stepping the particle beam across detection laser (and photomultiplier unit). Detection efficiency was calculated from photomultiplier counts of the optical detection stage normalized by external CPC or OPC concentration. Total particle transmission efficiency (TE) is obtained as a parameter of a fit function which results from a convolution of the assumed Gaussian particle beam profile and a rectangular function describing detection threshold.

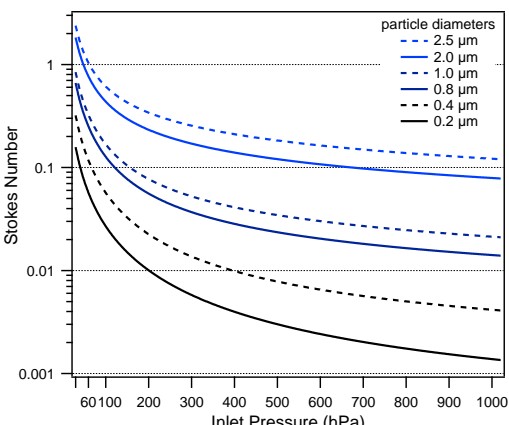

**Figure 7.** Stokes number calculated for different particle diameters as a function of inlet pressure in the range between 30 and 1020 hPa, with CPI behaviour of varying orifice diameter. Particle density of $1\,\mathrm{g\,cm^{-3}}$ is used comparable to polystyrene beads (which has $1.05\,\mathrm{g\,cm^{-3}}$). For other higher particles densities, the result can be upscaled proportionally.

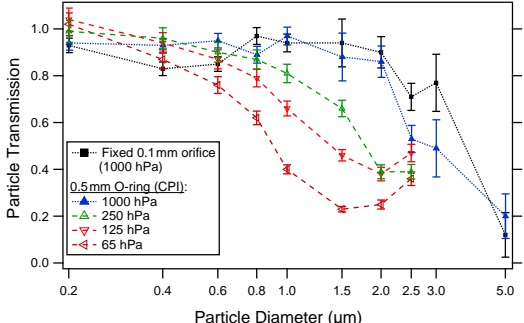

**Figure 8.** Transmission of PSL particles through the CPI device (blue line) and an aerodynamic lens in comparison to the same aerodynamic lens with a fixed 0.1 mm in diameter critical orifice at 1000 hPa (black line). The filled data markers denote results at ambient sea level pressure. Error bars include statistical particle counting uncertainty, as well as uncertainties of the fitted function parameters and tube loss corrections. Open markers denote results for inlet pressures of 250, 125 and 65 hPa. Measurements also were performed for 380 hPa but are not included here for clarity.



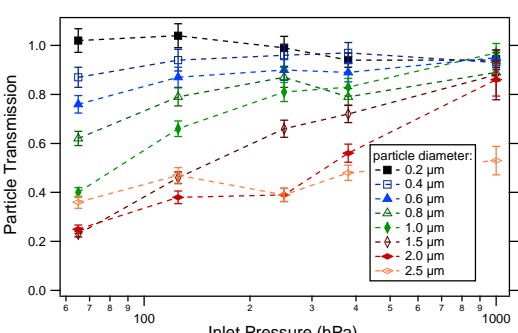

**Figure 9.** Transmission of PSL particles through the CPI device as a function of inlet pressure. Different PSL particle sizes from 0.2 to 2.5 μm were used in this experiment. Error bars include statistical particle counting uncertainty, corrections and uncertainty of maximum detection to transmission difference.