# Peer review of "Application of an O-ring pinch device as a constant pressure inlet (CPI) for airborne sampling"

_Atmospheric Measurement Techniques, 2020_

## Referee Comment (RC1) · Anonymous Referee #1 · 8 Apr 2020

This manuscript represents a thorough description and characterization of a new design for a constant pressure inlet for airborne aerosol and trace gas sampling applications. This design will benefit the community because it represents a significant improvement of constant pressure inlets on measurement platforms where small dimensions, automated operation and reliable performance in presence of rapid changes of ambient conditions are a requirement.

The manuscript is overall well written and concise. Some sentences could benefit from comments from a native language reader. The typesetting requires some streamlining of the use of italic vs roman fonts in formulae.

I would recommend the manuscript for publication in AMT after a few minor comments have been addressed.

Specific comments:

p2, l19: inner diameter x cross section (do not capitalize)

p2, l19: please give city/country information for reference to a manufacturer of equipment, no quotation marks.

p2, l23: include proper reference to the manufacturer

p2, l26: . . .O-ring, i.e., the tube cross section. . . (include commas and "the"). Consider rephrasing.

p2, l34: comma after "i.e."

p2, l35: ". . .not used for. . ." (d missing)

p2, l38: $D_{bore}$ – "bore" not in italics

p2, l45: include proper reference to the manufacturer of the pressure sensor

p2, l48: The research aircraft should be identified more consistently and clearly for readers outside the airborne science community (at least refer to the operating organizations of the respective platforms)

p3, l15: "meets" not "meet"

p3, l16: "particle" not "particles"

p3, sec 2 1st par: The discussion of particle losses is somewhat redundant to the next section where this is discussed in more detail. This paragraph motivating the use of visual inspection of the orifices could be shortened with reference to the next section. Consider moving this entire section after the discussion of transmission losses.

p3, 2nd par: Is there any information on longer-term stability of the results – do the O-rings degrade after a number of pinching cycles such that the particle transmission might change? How reproducible are those results with a different batch of O-rings made of the same material?

p3, l43: "Supposedly" seems to be the wrong word here.

p4, l8: include reference to manufacturer

p4, eq 1: erf should not be italicized

p4, l38: ...factor C, due to _the_ decrease... (missing "the". Parenthesis not really needed, include into previous sentence)

p5, l10: use roman font in formula subscripts "downstream" and "upstream".

p5, l42: identify research aircraft more clearly - see comment above.

p6, l2: "higher" -> "larger"

p6, sec 6: can the authors give an outlook to the performance of the CPI design for other particle measurements beyond aerosol mass spectrometry where the transmission characteristics of smaller particle sizes might be relevant?

Figure 1: Typo in "constant pressure inlet" Figure 7: the mixed use of color and line style is not very intuitive as only one parameter is varied here. Consider, e.g., using the same line style while labeling each line in the plot.

---

## Referee Comment (RC2) · Fred Brechtel (Referee) · 13 Apr 2020

Overall Comments

A novel, variable-diameter orifice inlet is described that facilitates the operation of airborne instruments requiring a constant inlet operating pressure. This is a useful work presenting a creative idea for actively pinching an Oring to create the variable orifice. Particle transmission efficiency study results validate successful operation of the new device.

I believe cloud condensation nucleus instruments also typically employ a constant pressure inlet for aircraft measurements. I suggest adding a sentence to the introduction referring to this application as it could benefit many readers.

[Figure]

Specific Comments

I have some concerns regarding the reproducibility of the circularity of the orifice diameter and how this might impact the particle transmission efficiency. The photos in Figure 5 are extremely useful toward understanding the behavior of the orifice diameter as a function of pinch. In Fig 5b the top 4 panels still appear to show non-circular orifice diameters. Please add a short discussion of the reproducibility of the transmission efficiency results for the same oring as well as after a new oring has been installed in the device.

Other technical questions that do not necessarily need to be addressed in the paper but would be interesting to understand include: expected lifetime of the oring, scheduled cleaning required due to collection of particles, ablation of oring material creating "rubber burrs" or altering the orifice circularity, oring fatigue due to constant pinching, and ozone exposure degrading the oring elasticity.

Technical Corrections

Page 2 line 7 I suggest changing to ". . .without additional pumping or bypass flow. . ."

Page 2 line 25 change to "The shape of the pinched orifice is critical toward avoiding significant. . ."

Page 2 line 36 do you mean 3.9 mm?

Page 2 line 45 is the range of the pressure sensor really only 0-10 hPa?

Page 2 line 49 can you comment on whether a +/-2% pressure deviation influences the flow enough that the transmission efficiency is affected?

Page 3 line 8-10 I suggest: "The pinching movement travel is limited by two optical sensors. One sensor prevents overloading the motor at maximum pinching while the second sensor prevents the mechanism from opening too far, which. . .."

Page 3 line 39 I suggest: "Laboratory tests with the 0.5mm Oring were performed prior

to field deployment during stratospheric flights. . ..”

Page 3 line 45 Would a more straight forward way to measure the transmission efficiency have been to operate an OPC/other particle detection instrument downstream of the aerodynamic lens while a similar instrument was measuring the particles entering the CPI/aerodynamic lens?

Page 4 Eqn 2 I believe particle diameter is supposed to be squared in the Stokes number relation.

Please review and rewrite the transmission efficiency test description at the bottom of page 4 and top of page 5 to make it clearer.

Page 5 line 50 “Nevertheless, even at the lowest inlet pressure of 65. . ..”

Please rewrite the top paragraph of page 6 to make it clearer.

Figure 1. Constant is spelled “Constatn” in the box in the figure

Figure 2. I would restate in the caption that the oring dimension 0.4x2.15 mm is Inner Diameter x Cross Section.

Figure 4. The caption: “. . .which differ in dynamic pressure” do you mean “. . .which differ by the dynamic pressure”?

Figure 8 I suggest making the fixed orifice results with a solid black line to make it easier to distinguish from the other curves. Why do the two lowest pressure curves shows an increase in TE at the largest particle sizes? Choose a different color for either the 125 hPa or 65 hPa results so they are easier to distinguish from each other.

Figure 9 caption: should it be: “Transmission of PSL particles through the CPI device and an aerodynamic lens as a function of. . .”

[Figure]

---

## Author Comment (AC1) · 20 May 2020

General Reply: We thank the Referee #1 for carefully reading the manuscript and finding many valid mistakes and suggestions for improvements. All those specific comments will be addressed. As suggested, we will ask a native language speaker for proofreading the manuscript.

Specific comments: p2, l19: inner diameter x cross section (do not capitalize)

Changed.

p2, l19: please give city/country information for reference to a manufacturer of equipment, no quotation marks.

[Figure]

Changed to: . . .Dichtelemente arcus GmbH/Germany.

p2, l23: include proper reference to the manufacturer

Changed to: . . .rubber from the company Zhermack SpA/Italy. . .

p2, l26: . . .O-ring, i.e., the tube cross section. . . (include commas and "the"). Consider rephrasing.

Rephrased: The outer diameter of the pinch O-ring, i.e. the tube's cross section was chosen to be relatively large, as it enables higher relative shrinking of the O-ring's inner opening.

p2, l34: comma after "i.e."

Corrected.

p2, l35: ". . .not used for. . ." (d missing)

Corrected.

p2, l38: D_bore – "bore" not in italics

Changed, also for the same cases above.

p2, l45: include proper reference to the manufacturer of the pressure sensor

Changed to: CMR373-Model, Pfeiffer Vacuum GmbH/Germany

p2, l48: The research aircraft should be identified more consistently and clearly for readers outside the airborne science community (at least refer to the operating organizations of the respective platforms)

Changed to: . . .(DLR-HALO G550, AWI-Polar 6 (DC-3), Myasishchev M-55 "Geophysica", NASA DC-8, DLR-Dassault Falcon 20).

p3, l15: "meets" not "meet"

Corrected.

p3, l16: "particle" not "particles"

Correctted.

p3, sec 2 1st par: The discussion of particle losses is somewhat redundant to the next section where this is discussed in more detail. This paragraph motivating the use of visual inspection of the orifices could be shortened with reference to the next section. Consider moving this entire section after the discussion of transmission losses.

We have slightly reduced the length of this section. However, we consider it as part of the engineering process for finding a good pinching geometry and O-ring dimensions, and thus we would not include it in the methods and results section. The high particle losses of the first O-rings are given here as motivation for a better solution. The transmission efficiency at this point was measured with a different technique, namely by flash vaporization (AMS Instrument) and often more qualitatively, since losses were too high anyway. The description of this method would add even more to the particle losses section. Given that there were a few iterations of the mechanical design and O-rings, we don't want to confuse the reader with different CPI versions and characterization techniques in the methods and results sections. For instance, an even softer silicon rubber material (shore hardness of 20) was tested.

p3, 2nd par: Is there any information on longer-term stability of the results – do the O-rings degrade after a number of pinching cycles such that the particle transmission might change? How reproducible are those results with a different batch of O-rings made of the same material?

Yes, this is a valid comment. Reviewer #2 has very similar questions and we comment on both remarks in Reply #2. We will also answer it in the revised manuscript.

p3, l43: "Supposedly" seems to be the wrong word here.

Changed to "Apparently"

p4, l8: include reference to manufacturer

Changed to: ... referenced by a CPC (Model 5.403, GRIMM Aerosol Technik/Germany) ...

p4, eq 1: erf should not be italicized

Corrected.

p4, l38: ...factor C, due to _the_ decrease. . . (missing "the". Parenthesis not really needed, include into previous sentence)

Changed to: On the other hand, a decrease of inlet pressure enhances the effects of other factors such as the growing Cunningham slip correction factor C, and the increasing flow speed in order to keep the mass flow constant.

p5, l10: use roman font in formula subscripts "downstream" and "upstream".

Corrected.

p5, l42: identify research aircraft more clearly - see comment above.

Changed as above.

p6, l2: "higher" -> "larger"

Corrected.

p6, sec 6: can the authors give an outlook to the performance of the CPI design for other particle measurements beyond aerosol mass spectrometry where the transmission characteristics of smaller particle sizes might be relevant?

We have added this to the summary: Furthermore, other particle counting instruments relying on constant pressure in their inlet or condensation cell, such as condensation particles counters (CPC, or cloud condensation nucleus counters, CCNC) can be equipped with the O-ring based CPI system. A CCN-200 (DMT, Longmont, CO, USA) was already deployed with the O-ring based CPI system (Andreae et al., 2018). However, it utilized a different O-ring with a controlled downstream pressure of about 200 hPa and an air flow of an order of magnitude higher. Therefore, this application requires a different study of the transmission efficiency.

Added to the summary: If used with a focus on even smaller particles sizes, in the nm size range, a high transmission efficiency is well maintained. This is attributed to the fact that impaction losses due to particles' smaller inertia are low, and diffusion losses are negligible due to the CPI's small internal volume and high flow rate.

Figure 1: Typo in "constant pressure inlet" Figure 7: the mixed use of color and line style is not very intuitive as only one parameter is varied here. Consider, e.g., using the same line style while labeling each line in the plot.

Corrected.

[Figure]

Stokes Number

particle diameters:

2.5 µm

2.0 µm

1.0 µm

0.8 µm

0.4 µm

0.2 µm

Inlet Pressure (hPa)

60 100 200 300 400 500 600 700 800 900 1000

0.1

0.01

0.001

**Fig. 1.** Figure 7

---

## Author Comment (AC2) · 20 May 2020

General Reply: We thank Fred Brechtel, Referee #2 for diligently reading the manuscript, finding mistakes and raising questions which can benefit the manuscript.

I believe cloud condensation nucleus instruments also typically employ a constant pressure inlet for aircraft measurements. I suggest adding a sentence to the introduction referring to this application as it could benefit many readers.

- Indeed, this can improve the visibility of our technique. This idea has been added to the abstract and the summary section. As requested by the first reviewer, we have also addressed the issue of smaller particle sizes. We modified it in the abstract to:

The CPI device can also be used in condensation particle counters (CPCs), cloud

condensation nucleus counters (CCNCs), and gas phase sampling instruments in a wide range of altitudes and inlet pressures.

Currently, one CPI device is already in use on a CCN-200 (DMT, CO, USA) instrument from the multiphase chemistry department of our institute. However, it utilizes a different O-ring with the controlled pressure of around 200 hPa and approximately 20 times higher flow. Added to the summary:

Furthermore, other particle counting instruments relying on constant pressure either in their inlet or condensation cell, such as condensation particles counters (CPCs, or cloud condensation nucleus counters, CCNCs) can be equipped with the O-ring based CPI system. A CCN-200 (DMT, Longmont, CO, USA) was already deployed with the O-ring based CPI system (Andreae et al., 2018). However, it utilized a different O-ring, with a controlled downstream pressure of about 200 hPa, and an air flow of an order of magnitude higher. Therefore, this application would require a different study of the transmission efficiency.

Specific Comments:

I have some concerns regarding the reproducibility of the circularity of the orifice diameter and how this might impact the particle transmission efficiency. The photos in Figure 5 are extremely useful toward understanding the behavior of the orifice diameter as a function of pinch. In Fig 5b the top 4 panels still appear to show non-circular orifice diameters. Please add a short discussion of the reproducibility of the transmission efficiency results for the same oring as well as after a new oring has been installed in the device.

- The production method of the O-ring is highly reproducible. The same mold was reused in the production of numerous O-rings, while the specification of the silicon rubber asserts a reproduction detail of 2 $\mu$m. A perfect circularity for the maximum pinching state (top 4 panels) is not possible. However, this does not appear as a prerequisite for good particle transmission. As stated in the manuscript and showed in

Figure 5, the smallest aperture always folded into a more triangular shape and worked well with respect to particle transmission. Also, with the same pinching mechanics, the maximum pinching states of a few O-rings from the same production batch showed exactly the same triangular deformation when observed under the microscope. The major factor to be avoided in the CPI design is the folding of the O-ring's aperture into bends along the flow axis. The exact reproducibility of the pinching is not mandatory, as long as the particle transmission quality remains as expected at ground level pressure (under strong pinching). This has to be confirmed by measurement after a new O-ring installation. With this considerations, further measurements at lower pressure are not necessary, because the O-ring relaxes towards its original, more circular shape. The particle transmission is limited by other (reproducible) factors, such as lower air density and the same fixed geometry for the flow. So far, a few O-rings made for other instruments of the research group showed very similar transmission results (TE > 85%) at sea level pressure for submicron particles and hence were kept for further usage.

The explanations above will be included in the revised manuscript.

Other technical questions that do not necessarily need to be addressed in the paper but would be interesting to understand include: expected lifetime of the oring, scheduled cleaning required due to collection of particles, ablation of oring material creating "rubber burrs" or altering the orifice circularity, oring fatigue due to constant pinching, and ozone exposure degrading the oring elasticity.

- The O-ring used in the transmission study of this manuscript was in use in the ERICA instrument for about two years. Because the instrument was sitting idle in the lab with a closed inlet for most of this time, the O-ring was also experiencing a maximum pinching state. It appears that the elasticity of the silicon rubber remains in good condition during the two year period, and no degradation of particle transmission with the same O-ring was found. Regarding ozone exposure, it can be added that silicone rubber is considered as one of few elastomers with very good ozone resistance. For instance, the same installed O-ring performed in two aircraft campaigns, one of which focused

on stratospheric measurements (Stratoclim campaign) with a flight duration of around 30 hours in the stratosphere. Cleaning of the O-ring was performed regularly, rather as a precaution or for troubleshooting different instrument problems. We do not recall any visible layering on the O-ring surface due to particle deposition. There was one case in which a larger dust piece obstructed the flow; it is likely that it might have fallen into the CPI during installation of an inlet line. For this reason, handling the O-ring and CPI's surroundings in clean conditions is preferable, but this applies to a fixed orifice setup as well. Traces of abrasion on the O-ring surface have not been discovered yet. This may be explained by the CPI's location, downstream of sampling line tubing, where abrasive super-micron particles are mostly lost.

Some text of the paragraph above will be included in the revised manuscript.

Technical Corrections:

Page 2 line 7 I suggest changing to ". . .without additional pumping or bypass flow. . ."

Changed as suggested.

Page 2 line 25 change to "The shape of the pinched orifice is critical toward avoiding significant. . ."

Changed as suggested.

Page 2 line 36 do you mean 3.9 mm?

No, this was correct. This Setup has a different, in-house made (quite large) aerodynamic lens, which is optimized for super-micron particles.

Page 2 line 45 is the range of the pressure sensor really only 0-10 hPa?

Yes, instead of using a sensor with a larger range of 0-100 hPa or 1000 hPa, the 10 hPa range provides best precision for a lens pressure controlled at a few hPa. The datasheet confirms a range of 10-3 to 11 hPa for linear voltage output limits, but we are not sure whether it benefits the reader to provide these details here.

Page 2 line 49 can you comment on whether a +/-2% pressure deviation influences the flow enough that the transmission efficiency is affected?

Yes, based on a laboratory test of transmission vs. lens pressure, with 350nm ammonium nitrate particles the estimated change of the transmission efficiency was about 2-3%. The set lens pressure of 4.5 hPa was chosen for maximum for particle transmission - the same pressure as that specified by the manufacturer. With respect to a few % of pressure difference, the transmission curve is smooth, as confirmed in the lab after the new lens was installed. The +/-2% is given with some safety margin. For instance, the maximum deviation for a flight shown in Figure 4 was only 1.2%. In practice, the lens pressure fluctuates equally above and below the set pressure, and short deviations in flow rate and transmission efficiency should average out over time. We expect this to be a negligible error as compared to other error sources of the whole system.

Page 3 line 8-10 I suggest: "The pinching movement travel is limited by two optical sensors. One sensor prevents overloading the motor at maximum pinching while the second sensor prevents the mechanism from opening too far, which. . .."

Yes, the "optical sensor" was unclear, it should be "optical distance sensor". There is only one optical sensor measuring the distance between the two moving parts. The sensor output values are used to limit the range of the motion. Rephrased to:

"The pinching movement has stop points at both travel ends. These limits are defined by the voltage output of an optical distance sensor measuring the distance to the pinching lever."

Page 3 line 39 I suggest: "Laboratory tests with the 0.5mm Oring were performed prior to field deployment during stratospheric flights. . .."

Changed as suggested.

Page 3 line 45 Would a more straight forward way to measure the transmission efficiency have been to operate an OPC/other particle detection instrument downstream of the aerodynamic lens while a similar instrument was measuring the particles entering the CPI/aerodynamic lens?

This would not be straightforward. Downstream of the aerodynamic lens, sufficiently good vacuum pressure (P<10-3 mbar) has to be maintained. Typical OPC instruments employ different techniques to define the active optical detection area (and flow rate), either by means of air flow restriction, e.g. with a sheath flow, or some optical focal plane discrimination. In theory, with a larger effort, one could redesign an existing OPC into the vacuum configuration by using only its optical detection part. However, we do not see a major difference or an improvement to just using the optical detection unit which we used in our setup. Rather, one would use the same optical detection unit from the vacuum chamber upstream of the aerodynamic lens. But at this position, the particle flow is not restricted to a narrow particle beam, and the problem of sample area/flow rate definition arises again; it has to be solved in similar ways as already done in existing OPC instruments.

Page 4 Eqn 2 I believe particle diameter is supposed to be squared in the Stokes number relation.

Yes, thank you very much for finding this error. In the calculation the equation was used correctly.

Please review and rewrite the transmission efficiency test description at the bottom of page 4 and top of page 5 to make it clearer.

Probably this paragraph was less clear because it was mixed with "anticipated" result statements which instead should be included in the results section. Those statements were redundant here and were removed.

Page 5 line 50 "Nevertheless, even at the lowest inlet pressure of 65. . .." Please rewrite the top paragraph of page 6 to make it clearer.

The sentences here were too long and laden with too many statements. This has been rewritten to: Nevertheless, even at the lowest designed and experienced inlet pressure of 65hPa, the CPI's particle transmission performance can be well-suited for stratospheric application where particle diameters are mostly below 0.6-0.8 $\mu$m. In cases of larger optical diameters, especially with respect to mean volume diameters (e.g. of fresh volcanic origin at stratospheric altitudes (Wilson et al., 1993)), one has to account for the evaporation of particle's water content and other volatile species in the inlet tubing upstream of the CPI. This effect leads to smaller aerodynamic diameters and effectively improves the particle transmission through the CPI, if one were judging by the ambient optical sizes.

Figure 1. Constant is spelled "Constatn" in the box in the figure

Corrected.

Figure 2. I would restate in the caption that the oring dimension 0.4x2.15 mm is Inner Diameter x Cross Section.

Changed as suggested to: . . .available O-rings of 0.8 $\times$ 2.0 mm (inner diameter $\times$ cross section),. . .

Figure 4. The caption: ". . .which differ in dynamic pressure" do you mean ". . .which differ by the dynamic pressure"?

Yes, corrected as suggested.

Figure 8 I suggest making the fixed orifice results with a solid black line to make it easier to distinguish from the other curves. Why do the two lowest pressure curves show an increase in TE at the largest particle sizes? Choose a different color for either the 125 hPa or 65 hPa results so they are easier to distinguish from each other.

- Good point, both lines for Fixed and CPI for 1000 hPa are now solid lines, albeit with different markers and colors.

- There is no obvious explanation for the increase in the TE for largest particle sizes. One explanation which would go in this direction is that the impaction losses on the expansion side, downstream of the O-Ring would be lower with the higher particle's inertia.

The colors for 125 and 65 hPa are changed to more distinct ones.

Figure 9 caption: should it be: "Transmission of PSL particles through the CPI device and an aerodynamic lens as a function of. . ."

Yes, this formulation is definitely better, same as in the figure caption before. Additionally, the line colors with respect to particle diameters were changed to the same as those in Figure 7.

Added citation:

Andreae, M. O., Afchine, A., Albrecht, R., Holanda, B. A., Artaxo, P., Barbosa, H. M. J., Borrmann, S., Cecchini, M. A., Costa, A., Dollner, M., Fütterer, D., Järvinen, E., Jurkat, T., Klimach, T., Konemann, T., Knote, C., Krämer, M., Krisna, T., Machado, L. A. T., Mertes, S., Minikin, A., Pöhlker, C., Pöhlker, M. L., Pöschl, U., Rosenfeld, D., Sauer, D., Schlager, H., Schnaiter, M., Schneider, J., Schulz, C., Spanu, A., Sperling, V. B., Voigt, C., Walser, A., Wang, J., Weinzierl, B., Wendisch, M., and Ziereis, H.: Aerosol characteristics and particle production in the upper troposphere over the Amazon Basin, Atmos. Chem. Phys., 18, 921–961, https://doi.org/10.5194/acp-18-921-2018, 2018.